# Study protocol for a randomised controlled trial: harmonising optimal strategy for treatment of coronary artery stenosis — coronary intervention with next-generation drug-eluting stent platforms and abbreviated dual antiplatelet therapy (HOST-IDEA) trial

Chi-Hoon Kim,[1] Jung-Kyu Han,[1] Han-Mo Yang,[1] Kyung Woo Park,[1] Hae-Young Lee,[1] Hyun-Jae Kang,[1] Bon-Kwon Koo,[1] Namho Lee,[2] Tae-Joon Cha,[3] Tae-Hyun Yang,[4] Myung-Ho Jeong,[5] Myeong-Ho Yoon,[6] Seung Uk Lee,[7] Seung Jin Lee,[8] Jin Won Kim,[9] Jin-Man Cho,[10] Kyu-Rock Han,[11] Wook Bum Pyun,[12] Hyo-Soo Kim[1]

C-HK and J-KH contributed equally.

For numbered affiliations see end of article.

Correspondence to
Dr Hyo-Soo Kim;
hyosoo@snu.ac.kr

## ABSTRACT

**Introduction** We have recently seen the introduction of newer generation drug-eluting stents with ultrathin struts that use advanced polymer technologies. However, the efficacy and safety of these newest stents have not yet been fully explored. In addition, there are still controversies over the optimal duration of dual antiplatelet therapy (DAPT) after stent implantation, particularly for ultrathin stents with the newest polymer technologies.

**Methods and analysis** The HOST-IDEA trial is a randomised, open-label, multicentre, non-inferiority trial and the first study to directly compare two of these ultrathin sirolimus-eluting stents: Orsiro stent with biodegradable polymer, and polymer-free Coroflex ISAR (CX-ISAR) stent. This study has a scheme of 2×2 factorial design according to the stent type and DAPT duration (3 vs 12 months). A total of 2152 patients will be randomised and stratified to demonstrate the non-inferiority of CX-ISAR to Orsiro, or of the abbreviated DAPT duration to the conventional 12 months (both in 1:1 ratio). For the comparison of stent type, the primary endpoint is target lesion failure (TLF), which is a composite of cardiac death, target vessel-related myocardial infarction and clinically driven target lesion revascularisation. For the comparison of DAPT duration, the net adverse clinical event is the coprimary endpoint, which is defined as a composite of TLF, definite/probable stent thrombosis and major bleeding.

**Ethic approval and dissemination** All the institutions involved in this study are required to have ethical approval prior to patient enrolment. This multicentre study will recruit patients through competitive registration, but institutions that have not yet obtained ethical approvals have made it impossible to enrol patients in a centralised web database. The final results will be presented at relevant international conferences and will be materialised in the form of papers.

## Strengths and limitations of this study

► To the best of our knowledge, the clinical outcome of two up-to-date coronary stents with ultrathin strut, the Orsiro and Coroflex ISAR stents will be first compared in this randomised clinical trial. These two stents are based on the latest drug coating technology, however, no randomised studies have been reported to directly compare these two stents.

► We could derive a meaningful result on the optimal duration of dual antiplatelet therapy (DAPT) in stents using ultrathin strut by adopting a 2×2 factorial design with a difference in the duration of DAPT maintenance (3- vs. 12 month). This study will confirm that the clinical performance is not worse than the conventional 12 month maintenance even if the DAPT maintenance period is kept short in the latest stents with thin strut thickness.

► We also will be able to simultaneously test the difference between two co-primary outcomes, target lesion failure and net adverse clinical outcome, since we will register sufficient numbers of patients to secure statistical power.

► To minimise potential risks and ensure patient safety, patients with ST-segment myocardial infarction (STEMI) who are generally recommended to apply a DAPT maintenance period of 1 year or more will be excluded from this study.

**Trial registration number** NCT02601157; Pre-results.

## INTRODUCTION

Second generation drug-eluting stents (DESs) has been introduced to overcome the limitations of earlier versions of DESs such as late stent thrombosis and late catch-up.[1] The improvements of second generation DESs were made in many different fields as follows: better stent design with greater conformability, thinner strut thickness by using of new metal alloy, optimal load and improved release kinetics of drugs, and new polymer technology. All these improvements made second generation DESs safer and more efficacious.[2 3]

Thin strut thickness makes greater conformability, better deliverability, and lesser injury. This creates lesser shear disturbances, and reduces peri-strut inflammation and fibrin deposition, finally contributing to improved re-endothelialization.[4] Currently, the Orsiro hybrid siro-limus-eluting stent (SES) (Orsiro, Biotronik AG, Bülach, Switzerland) and the Coroflex ISAR (CX-ISAR, B. Braun Melsungen AG, Berlin, Germany) SES are two commercially available stents with thinnest strut thickness (60 μm for diameter ≤3.0 mm, 80 μm for diameter ≥3.5 mm for Orsiro, 50 μm for diameter ≤2.5 mm, 60 μm for diameter ≥2.75 mm for CX-ISAR). Interestingly, these two stent systems adopt two different up-to-date drug coating technologies. The Orsiro utilises poly-L-lactic acid (PLLA) for biodegradable polymer.[5] While, sirolimus of the ISAR platform is coated on the stent strut without any polymer substance.[6]

The Orsiro stent showed non-inferior safety and efficacy outcomes compared with everolimus-eluting stents (EESs).[7] In contrast, previous version of the ISAR stent with stainless steel backbone was compared with zotaro-limus-eluting stent (ZES), demonstrating its non-inferior safety and efficacy.[8] However, CX-ISAR, a latest version of the ISAR system with cobalt chromium (CoCr) alloy backbone, has not yet been tested in a large scale randomised controlled trial. Furthermore, up to now, there is no head-to-head comparison between these two ultrathin stents with distinct drug eluting technologies, the Orsiro and CX-ISAR.

Meanwhile, optimal duration of dual antiplatelet therapy (DAPT) after DES implantation is still unconcluded. Current guidelines recommend 6–12 months of DAPT after DES implantation for patients with stable angina.[9] However, there are controversial studies regarding longer- versus shorter-duration DAPT. In particular, we have no data regarding optimal duration of DAPT for ultrathin stents with advanced polymer technologies. Because of potential of better re-endothelialization and less polymer-related adverse effects,[1 4] we can reasonably guess that shortened DAPT would be enough for these newer generation stents.

Therefore, this prospective, randomised, open-label, 2×2 factorial design multicenter trial was planned to address: (1) whether newly-developed ultrathin stents (Orsiro, CX-ISAR) are comparable to each other in terms of efficacy and safety, and (2) whether short duration of DAPT is non-inferior to conventional 1 year duration in patients receiving ultrathin newer generation DESs.

## METHODS AND DESIGN

### Study design and primary hypothesis

The HOST-IDEA trial is a randomised, open-label, multicenter, non-inferiority trial comparing the Orsiro with the CX-ISAR. The trial is powered to investigate a hypothesis that a polymer-free stent platform (CX-ISAR) is non-inferior to a biodegradable polymer-based stent (Orsiro) as regards post-procedure 1 year target lesion failure (TLF), as a composite of cardiac death, target vessel myocardial infarction (TVMI), and clinically driven target lesion revascularisation (TLR). At the same time, another hypothesis will also be examined: 3 months' DAPT may deliver the same clinical efficacy and safety as conventional 1 year DAPT strategy. For this purpose, net adverse clinical events (NACE), defined as a composite of TLF, definite or probable stent thrombosis, and major bleeding according to the pertinent criteria,[10 11] will be checked as a co-primary endpoint. A 2×2 factorial design will be used to address these questions. Unless there is significant interaction between the two interventions, this factorial design can provide a useful scheme for testing two interventions simultaneously in a single dataset, and can be used to minimise sample size without limiting the statistical power.[12]

### Study population and eligibility criteria

All participating centres are tertiary referral hospitals in Korea. Patients eligible for coronary intervention will be qualified with coronary angiography before the enrollment. Every participant will have at least one stenotic coronary lesion of >50% diameter stenosis suitable for stent implantation. To secure the statistical power and to obtain clear practical implications, high-risk patients for ischaemic adverse events will be excluded in this protocol; patients with ST-segment elevation myocardial infarction (STEMI) or unstable conditions such as cardiogenic shock or severe heart failure at the time of presentation. Detailed criteria for inclusion and exclusion are listed in box 1.

### Rationale for sample size calculation

Among the contemporary DESs, no stent platform has been directly compared with both Orsiro and CX-ISAR. And, the latest version of the ISAR system with a CoCr backbone, CX-ISAR, has not yet been tested in a large-scale randomised controlled trial (RCT). Instead, the Orsiro and previous versions of the ISAR system have been tested against EES and ZES, respectively, and clinical outcomes were found to be comparable in a number of large-scale studies.[7 8 13 14] The Orsiro was non-inferior to the Xience EES in the BIOSCIENCE trial,[7] and previous stainless steel-based ISAR platform and the Resolute ZES

**Box 1  Inclusion and exclusion criteria for the HOST-IDEA trial**

**Inclusion criteria**
► Patients with de novo stenotic lesions who are suitable for coronary stenting with drug-eluting stent

**Exclusion criteria**
► High risk profiles for ischaemic adverse events such as
  – ST-segment elevation myocardial infarction (STEMI)
  – Patients with cardiogenic shock or concomitant severe decompensated heart failure
  – Patients with myocardial infarction or stent thrombosis in spite of the maintenance of antiplatelet therapy
  – Restenosis in stented segments or previous sites of balloon angioplasty
► Patients who cannot follow allocated DAPT schedule due to the planned surgery or elective procedure within 3 months after the stenting
► Recent history of major surgery or evident events of gastrointestinal bleeding within 1 month from the procedure
► Patients on anticoagulation therapy with warfarin or other anticoagulants
► Life expectancy less than 1 year (such as malignancies or other chronic systemic diseases)
► Pregnant women
► Past history of allergy or other contraindications for the following medications/materials: aspirin, clopidogrel, prasugrel, ticagrelor, heparin, cobalt chromium, sirolimus

system had similar efficacy in the ISAR-TEST five trial.[8] As ZES and EES have similar efficacy in treating coronary disease,[15 16] it is reasonable to assume that the Orsiro and ISAR system will have similar levels of clinical efficacy and safety.

Other evidence supports this assumption. The TLF rate of Orsiro in the BIOSCIENCE trial was 6.5% over a 1 year follow-up.[7] In contrast, the ISAR system had a 1 year TLF rate of 13.1% in the ISAR-TEST five study.[8] This discrepancy was mainly due to the difference in TLR, rather than cardiac death or TVMI. The TLR rate of the ISAR system was somewhat higher than that of the Orsiro (1.9% cardiac deaths in the Orsiro vs. 1.9% in the ISAR system, TVMI 2.9% vs 2.4%, TLR 4.0% vs 10.3%, respectively). However, this contrast may be due to the study's policy regarding angiographic follow-up rather than the nature of the stent system itself. In the ISAR-TEST five trial, 6–8 months after the procedure, about three-quarters of patients (76.3%) had undergone dedicated angiographic surveillance,[8] whereas in the BIOSCIENCE trial, angiography was performed at the time of 13 month follow-up. Consequently, 1 year clinical outcomes of the Orsiro could avoid the potential influence of routine angiographic follow-up, and this trial was able to minimise the risk of oculostenotic revascularisation for non-ischaemic intermediate lesions.[17] In a similar vein, 1 year TLR rate of 10.4% for the Resolute ZES system was also reported in the ISAR-TEST five trial, but in the Resolute all-comer study, without the impact of routine angiographic

follow-up within 12 months, the same stent system had a lower TLR rate of 3.9%.[15]

Based on these data, it is reasonable to assume that the CX-ISAR and Orsiro would have similar TLR rates. Accordingly, 1 year TLF rate of the Orsiro in the BIOSCIENCE trial was employed as a reference for power calculation. With the assumption of a TLF rate of 6.5% and allowing for about 10% withdrawals or dropouts, a total of 2152 patients in 1:1 randomization will provide more than 80% power to detect a non-inferiority margin of 2.8% with a one-sided type I error of 0.05. These parameters are comparable to those of the BIOSCIENCE trial (reference value and non-inferiority margin 8.0% and 3.5%, respectively).[7] This size calculation may be able to secure the statistical power in case the event rates may be lower than expected. Non-inferiority margin of 2.5% for event rate of 5%, or non-inferior margin of 2.7% for event rate of 6% can be examined with this sample size even allowing for 10% withdrawal or dropouts.

This sample size might also be sufficient to test the second hypothesis. To date, no detailed data on the NACE rate of the CX-ISAR system have been reported. For the Orsiro stent, because most cases of stent thrombosis can also be counted as TLF events (cardiac death or MI), rates of 6.5% for TLF and 3.0% for major bleeding could be cited as references for 1 year NACE rate. Assuming patients with the Orsiro and 1 year DAPT have a NACE rate of 9.5%, 1039 patients will be required for each group of 3 months vs. 1 year DAPT to distinguish a 3.2% margin of non-inferiority with an α value of 0.05% and 80% power. There is no actual interaction between the two interventions of randomization, maintenance duration of DAPT and allocated stent types. Therefore, even though the sample size of 2152 patients might be insufficient to tell the difference of individual components of NACE, this sample size might cover some patient loss and could provide enough statistical power to verify the second hypothesis simultaneously in a 2×2 factorial design. The 3.2% non-inferiority margin is similar to the reference values of the RESET and OPTIMIZE trials.[18 19]

### Enrollment, procedure and study medications

After the identification of the target lesion and patients' consent to participate, randomization to stent type and DAPT duration will be performed using an electronic web-based database (© Cardiovascular Centre, Seoul National University Hospital, and CRSCube Software, Seoul, South Korea) according to the 2×2 factorial design (figure 1). Block randomization with block size of 8 and equal allocation probability for each group will be maintained throughout the entire study period. To ensure the randomization more secure, we have set a small block size for this study, and have not stratified by each participating centre. All the information about demographic, procedural, and follow-up data will be integrated into this centralised encrypted electronic database. Though this trial is an open-label study, these data will be managed

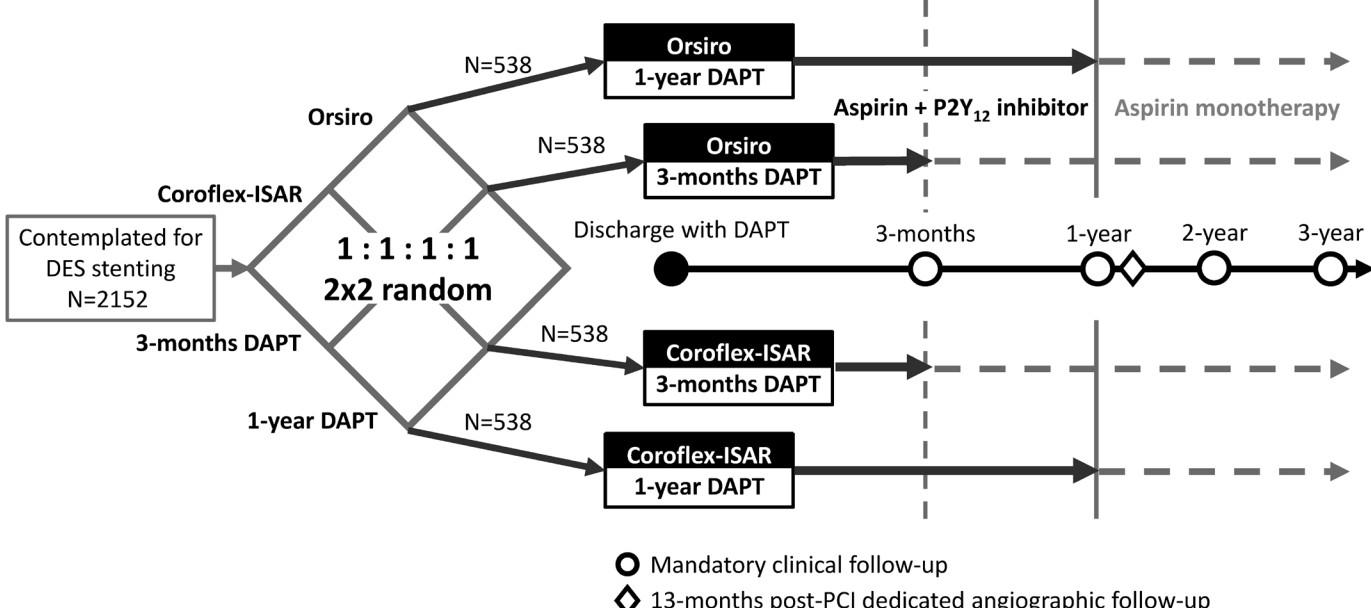

**Figure 1** Study outline and randomization scheme.

by independent research nurses or other well-trained professionals.

Coronary intervention will be performed according to generally accepted current guidelines.[9 20] To improve applicability of this trial, DAPT regimen with prasugrel or ticagrelor as well as clopidogrel will be allowed. Every antiplatelet-naïve patient undergoing an elective procedure will be given 300 mg aspirin and loading dose of one of P2Y$_{12}$ receptor inhibitors (eg, 600 mg clopidogrel, 60 mg prasugrel or 180 mg ticagrelor) preferably ≥2 hours before the intervention. These loading doses can be waived for chronic antiplatelet users. Choice for P2Y$_{12}$ inhibitors will be based on the current guidelines as well as patient/lesional characteristics. To avoid possible bias, clopidogrel will be preferentially used for patients with stable angina whereas prasugrel or ticagrelor will be used mainly for patients with acute coronary syndrome.[9] Patients with higher bleeding risk such as patients older than 75 years of age, history of ischaemic stroke or propensity to bleed can be treated with clopidogrel even in the case of acute coronary syndrome.

During the procedure, unfractionated heparin in a dose of at least 5000 IU or 70–100 IU/kg body weight will be administered for anticoagulation, while bail-out glycoprotein IIb/IIIa inhibitors, such as abciximab, will be left to the operators' discretion. Any lesional characteristics will be allowed for enrollment, except for in-stent restenosis of stented segments or previous treatment with balloon angioplasty. In addition to the angiographic findings, additional evaluations, such as intravascular ultrasound, optical coherence tomography (OCT), or fractional flow reserve assessed with pressure wire, may be used to characterise target lesions. Staged intervention can also be reserved for patients with complex lesional characteristics or multivessel disease, but even in these cases, target lesions can be treated only with the allocated stent platform. Because the Orsiro and CX-ISAR systems have similar configuration (see appendix), they are expected to be interchangeable, with no substantial differences in the procedure.

## Follow-up and data collection

A DAPT schedule according to the web-based randomization is mandatory for every participant. Daily maintenance dose of aspirin is 100 mg and all patients will be given maintenance dose of P2Y$_{12}$ inhibitors according to their allocation (clopidogrel at a dose of 75 mg daily, prasugrel 10 mg daily (5 mg daily for patients with body weight of less than 60 kg) or ticagrelor 90 mg twice a day). DAPT will be continued for up to 3 months or 1 year, as planned. To check patient's adherence to the DAPT regimen, a drug compliance survey will be conducted at the 1- and 3 month clinical follow-ups. Within this initial 3 months of follow-up, any unexpected discontinuation of DAPT will be regarded as a violation of the study protocol. After 3 month follow-up, brief interruptions of ≤5 days for elective surgery or planned procedures will be permitted, but every interruption of ≥6 days will be classified as non-adherence to the allocated 1 year DAPT regimen. This issue will be addressed for every case during the follow-up. Clinical follow-ups at the time of 1-, 3- and 12 month after the study enrollment are mandatory for the completion of the study. And during the period between 3 and 12 months, every patient will be required for out-patient visits at an interval less than 3 months. In addition to survey for DAPT compliance at the time of every out-patient visit, telephone interview will be given to patients who miss their scheduled appointments. DAPT with aspirin and clopidogrel/prasugrel/ticagrelor may be also extended beyond the predefined period, according to the

patient's risk and the responsible clinician's discretion. Allocated P2Y$_{12}$ inhibitors will not be changed to other agents during the entire study period. But for patients with higher ischaemic risk, prasugrel or ticagrelor may replace clopidogrel after the predefined period of DAPT maintenance.

Regardless of violation or drop-out, clinical outcomes as well as drug compliance up to 3 years will be collected in the centralised web database; data entry will be assigned to independent professionals. Any serious adverse events, including death, MI, revascularisation, stent thrombosis, and bleeding will be entered into the web database for up to 3 years in a blinded fashion. These events will be adjudicated independently by a blinded adjudication committee. Central and on-site data monitoring will be performed according to a predefined monitoring plan. Every electronic case report form will be checked by central data monitoring. On-site monitoring will also be performed to secure data integrity; records of the first 10 patients and subsequently a random 20% of the total registered patients will be verified. Dedicated angiographic surveillance will be scheduled at the time of 13 month follow-up, but this is not mandatory. Detailed instructions will be provided to each institution.

### Statistical analysis

Interim analyses will be performed to test the feasibility of this trial when half of the enrolled patients have their 1 year results. Potential interactions between the stent types and the recommended maintenance duration of DAPT will be identified to ensure the statistical power of this trial, before the study hypotheses are addressed.[12] The primary outcome will be examined from an intention-to-treat viewpoint, but considering the potential influences of protocol violation or drop-out, per-protocol analysis will be used at the same time. The per-protocol population will be limited to patients with (1) a successful procedure treated solely with the allocated stent type, (2) no violation of recommended antiplatelet strategy, and (3) complete clinical follow-up information.

Using the proportional-hazards model, clinical outcomes will be compared between the stent types and DAPT strategies, possibly after controlling for relevant covariates. Stratified analyses for the primary endpoint across major subgroups will be performed using the Mantel–Cox method. Subgroup analyses will be stratified by male sex, smoking history, diabetes mellitus, chronic kidney disease of stage ≥3, off- versus on-label indication, small vessel (≤2.75 mm) or long target lesion (>28 mm), and complex lesion (type B2/C) or chronic total obstruction. Rates of bleeding complications will be analysed according to the allocated antiplatelet strategy.

### Definitions of outcome

Outcome measures and endpoint concepts will follow the definitions suggested in current recommendations.[11] The primary endpoint in this trial is TLF, as defined above, while the composite outcome of NACE will be managed as a co-primary endpoint. Detailed definitions are summarised in the appendix.

### DISCUSSION

The HOST-IDEA RCT is the first study to directly compare two ultrathin CoCr backbone stents with up-to-date polymer technologies: the sirolimus-eluting Orsiro stent with biodegradable polymer, and the sirolimus-eluting polymer-free CX-ISAR stent. These two stents are clearly distinct from other contemporary drug-eluting stents (DES) with durable polymers. Strut thickness is 50–60 μm for Orsiro stent, and 60–80 μm for CX-ISAR stent. Ultrathin strut thickness makes these stents more conformable and deliverable. In addition, injuries to the arterial wall during stent implantation and subsequent peri-strut inflammation can be minimised.[4] This feature may also contribute to more rapid arterial healing and more reliable endothelial coverage.[13 21] Further, these two stent platforms have adopted different up-to-date polymer technologies. Orsiro stent utilises two-tiered hybrid polymer coating technology.[13 22] The outer layer is made of poly-L-lactic acid, which completely degrades over about 1 year period. The inner layer is silicon carbide inert matrix, which prevents the CoCr struts from being exposed to the diseased segment.[5] This unique hybrid system may greatly reduce chronic local inflammation around the stent struts and lessen the risk of denuded struts without re-endothelialization. In contrast, CX-ISAR utilises polymer-free drug release technology. A mixture of sirolimus, probucol, and shellac resin is mounted into numerous micropores on the stent strut.[6 23] Since the sirolimus in this dual drug-delivery system has the same eluting profile to that of the lipophilic solvent probucol, controlled drug release is enabled for up to 6–8 weeks, and nothing will be left on the stent struts after 3 months.

Few previous studies assessed the clinical outcome of the Orsiro stent. The BIOSCIENCE trial randomised 1063 patients to Orsiro and 1056 patients to Xience EES stent.[7] Clinical efficacy of the Orsiro stent was comparable to that of EES, widely used durable polymer stent (1 year TLF rate in Orsiro 6.5% vs 6.6% in EES group, P for non-inferiority <0.0004). The safety profile of the Orsiro was also reliable: only nine cases (0.9%) were reported as definite stent thrombosis during the 1 year follow-up period, compared with four cases (0.4%) in EES group (p=0.16). Interestingly, in this trial, STEMI patients treated with the Orsiro were associated with a lower risk for 1 year TLF: seven cases (3.3%) of 211 STEMI patients in Orsiro group vs. 17 (8.7%) of 196 in EES group (rate ratio 0.38, 95% CI 0.16 to 0.91, P for interaction 0.014). In contrast, the efficacy and safety of CX-ISAR have not been examined yet in a large-scale RCT. The HOST-IDEA RCT will provide the data regarding the efficacy and safety of the Orsiro and CX-ISAR by comparing these two stents.

In addition, there is still controversy over the optimal duration of DAPT following DES implantation. Several previous trials demonstrated that a shortened DAPT

strategy was comparable to conventional 1 year strategy. The REAL-LATE/ZEST-LATE trials analysed 2701 patients who had received SES (57%), paclitaxel-eluting stent (PES) (24%), ZES (19%) or other DESs.[24] The results demonstrated that DAPT longer than 12 months was not more effective compared with aspirin monotherapy to reduce the rate of myocardial infarction (MI) or cardiac death. Our group previously compared 6 month to 12 month DAPT in 1443 patients who underwent Xience/Promus EES or Cipher SES implantation in the EXCELLENT RCT.[25] We revealed that 6 month DAPT did not increase the risk of target vessel failure or the incidence of safety endpoint. And the RESET trial showed that 3 month DAPT following Endeavour ZES implantation was non-inferior to 12 month DAPT following other DES in 2117 patients with respect to the occurrence of the primary endpoint consisting of cardiovascular death, MI, stent thrombosis, target/vessel revascularisation, or bleeding.[18] In the PRODIGY trial, 2013 patients who received bare-metal stent, PES or EES were randomly allocated to take 6 month or 24 month DAPT. As a result, a 24 month DAPT was not more effective than 6 month regimen in reducing the composite of all-cause mortality, MI or cerebrovascular accident, whereas there was a greater risk of bleeding in the 24 month group.[26] In the OPTIMIZE trial including 3119 patients after ZES implantation, 3 month DAPT was non-inferior to 12 month therapy for the primary endpoint composed of all-cause mortality, MI, stroke, or major bleeding.[19] In contrast, some recent trials demonstrated that prolonged DAPT significantly reduced thrombotic adverse clinical events. In the DAPT trial, 9961 patients after 12 month DAPT following DES implantation were randomly assigned to continue DAPT or not. DAPT beyond 1 year after DES placement reduced the risks of stent thrombosis and major adverse cardiovascular and cerebrovascular events at the expense of an increased risk of bleeding.[27] The PEGASUS-TIMI 54 trial randomly allocated 21 162 patients with MI more than 1 year earlier to take an additional ticagrelor or placebo on top of aspirin. The results showed that prolonged DAPT using ticagrelor reduced the risk of cardiovascular death, MI, or stroke, but increased rates of major bleeding.[28]

Regarding this DAPT issue, the HOST-IDEA RCT also examines the safety and efficacy of the abbreviated DAPT duration in DAPT duration arm. Particularly, this trial will provide the first evidence regarding the optimal DAPT duration for the ultrathin stents with the newest polymer technologies. In addition, this study may also provide meaningful data on the clinical usefulness of the 3 month DAPT regimen for small-vessel intervention. In fact, the intervention for small-vessel diameter (<3 mm) is an item of the DAPT score and is included in the model to predict the occurrence of the future ischaemic adverse events.[29] In this regard, the recent guideline has advised that long-term DAPT maintenance should be considered for small-vessel intervention.[30] But, there is data that contradict this recommendation. Some previous studies such

as the RESET and OPTIMIZE trials have shown that the 3 month DAPT regimen is clinically useful and safe even in small-vessel intervention.[18 19] However, these studies used the Endeavour and Resolute zotarolimus-eluting stents, which are no longer used in the current clinical practice. To date, detailed data on the effect of the combination of the 3 month DAPT regimen and the new stent platforms on small-vessel intervention are very scarce. In this context, even if the 3 month DAPT regimen were used for the Orsiro or the CX-ISAR stents, we expect that excellent clinical outcome can also be achieved in small-vessel intervention.

## ETHICAL APPROVAL AND STATUS OF THE TRIAL
### Ethics approval and consent to participate
At the time of submission (February 2017), a total of 12 centres are participating in this trial. This prospective trial had been approved from eight centres, as of February 2017, name of the regulation authority and the issued IRB number are as follows: the review board of Seoul National University Hospital (D-1508-118-697), SoonChunHyang University Cheonan Hospital (SCHCA 2016-03-012), Hallym University Kangnam Sacred Heart Hospital (2016-04-31), Korea University Guro Hospital (MD16043), Ajou University Hospital (AJIRB-MED-DE4-16-170), Chonnam National University Hospital (CNUH-2016–096), Kwangju Christian Hospital (KCH-D-2016-03-003), Inje University Busan Paik Hospital (16-0117). Other four centres are on the review process with the same protocol and consent form (Kosin University Gospel Hospital, Kyung Hee University Hospital at Gangdong, Ewha Womans University Medical Centre Mokdong Hospital, Hallym University Kangdong Sacred Heart Hospital). Recruitment will not begin in any of these four centres until all local approvals have been obtained. The steering committee of this trial takes the responsibility for the study design. 12 independent regulation authorities of this trial are in full compliance with Good Clinical Practice as defined under the Korean Ministry of Food and Drug Safety regulations and the International Conference on Harmonisation guidelines.

All patients will receive sufficient information to make a decision about participation before providing their written informed consent. Informed consent will be obtained by independent research nurses of well-trained personnel of each participating centre. And every participant will have the right to withdraw their consent without restriction. Deferred consent will not be permitted for this study. Consent to publication will be obtained as a part of the general consent form, and individual patient data will be processed anonymously.

### Trial registration and current status of this trial
This trial was registered at clinicaltrials.gov (NCT02601157; November 7, 2015). On January 28, 2016, we enrolled patients for the first time at the coordinating centre (Seoul National University Hospital), since then

six participating centres have begun to register patients. A total of 143 patients had been enrolled as of February 2017,[25] we expect patient registration will be extended to other institutions by the end of this year. This paper translates the study protocol version 1.0. Any protocol amendments or revisions will be communicated with researchers involved in this trial and mentioned in the results paper.

## Availability of data and material

 But, study results and conclusion of this trial will be separately covered in the results paper, and the dataset of this trial will not be shared unless otherwise stated.

## Author affiliations

[1]Cardiovascular Centre, Department of Internal Medicine and Division of Cardiology, Seoul National University Hospital, Seoul, Korea
[2]Department of Internal Medicine and Division of Cardiology, Hallym University Kangnam Sacred Heart Hospital, Seoul, Korea
[3]Department of Internal Medicine and Division of Cardiology, Kosin University Gospel Hospital, Busan, Korea
[4]Department of Internal Medicine and Division of Cardiology, Inje University Busan Paik Hospital, Busan, Korea
[5]Department of Internal Medicine and Division of Cardiology, Chonnam National University Hospital, Gwangju, Korea
[6]Department of Internal Medicine and Division of Cardiology, Ajou University Hospital, Suwon, Korea
[7]Cardiovascular Centre, Kwangju Christian Hospital, Gwangju, Korea
[8]Department of Internal Medicine and Division of Cardiology, Soon Chun Hyang University Cheonan Hospital, Cheonan, Korea
[9]Department of Internal Medicine and Division of Cardiology, Korea University Guro Hospital, Seoul, Korea
[10]Department of Internal Medicine and Division of Cardiology, Kyung Hee University Hospital at Gangdong, Seoul, Korea
[11]Department of Internal Medicine and Division of Cardiology, Hallym University Kangdong Sacred Hospital, Seoul, Korea
[12]Department of Internal Medicine and Division of Cardiology, Ewha Womans University Medical Center Mokdong Hospital, Seoul, Korea

**Contributor** As described in manuscript.

**Contributors** H-SK, as a corresponding author, proposed the original concept and idea for this HOST-IDEA trial and supervised overall process of preparation. C-H K and J-K H made a draft of the study protocol and prepared this manuscript (these two authors equally contributed to this paper.) Patient consent form and centralised web-based database were prepared by C-HK. H-MY, KWP, H-YL, H-JK and B-KK critically reviewed the study protocol and this manuscript, and contributed to the overall trial design. In particular, these coauthors made significant contributions on the design of web-based database and approved the final version of the database. NL, T-JC, T-HY, M-HJ, M-HY, SUL, SJL, JWK, J-MC, K-RH and WBP reviewed the study protocol and proposed helpful ideas and information, and these coauthors are in charge of registering the trial protocol to the review boards of each participating institution. Final manuscript of this paper was reviewed and approved by all authors.

**Funding** This study was supported by the Korea Health Technology R&D Project (the grant from the 'Korea research-driven hospital' project, HI14C1277) through the Korea Health Industry Development Institute (KHIDI), funded by the Ministry of Health & Welfare (MHW). And this study is an investigator-initiated trial, which is partly supported by an unrestricted grant from Biotronik Korea (Seongnam si, Gyeonggi-do, South Korea) and B Braun Korea (Seoul, South Korea). These two companies equally contributed to this study, but funding source did not involve the study design. And they will have no role in the data collection and management. Data analysis, its interpretation and publication will be done without any interference.

**Competing interests** None declared.

**Patient consent** Detail has been removed from this case description/these case descriptions to ensure anonymity. The editors and reviewers have seen the detailed information available and are satisfied that the information backs up the case the authors are making.

**Ethics approval** Institutional Review Board of Seoul National University Hospital.

**Provenance and peer review** Not commissioned; externally peer reviewed.

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
