## [Reviewer comments · BMJ Open]

ARTICLE DETAILS

TITLE (PROVISIONAL)	Study protocol for a randomized controlled trial: Harmonizing Optimal Strategy for Treatment of coronary artery stenosis – coronary Intervention with next generation Drug-Eluting stent platforms and Abbreviated dual antiplatelet therapy (HOST-IDEA) trial
AUTHORS	Kim, Chi-Hoon; Han, Jung-Kyu; Yang, Han-Mo; Park, Kyung Woo; Lee, Hae-Young; Kang, Hyun-Jae; Koo, Bon-Kwon; Lee, Namho; Cha, Tae-Joon; Yang, Tae-Hyun; Jeong, Myung-Ho; Yoon, Myeong-Ho; Lee, Seung Uk; Lee, Seung Jin; Kim, Jin Won; Cho, Jin-Man; Han, Kyu-Rock; Pyun, Wook Bum; Kim, Hyo-Soo

VERSION 1 - REVIEW

REVIEWER	Gen-Min Lin Hualien-Armed Forces General Hospital, Taiwan
REVIEW RETURNED	08-Apr-2017

GENERAL COMMENTS	Dr. Kim and colleagues will design a RCT, 2x2 factorial, non-inferior trial to compare TLF and DAPT between Orsiro and CX-ISAR. In general, the study is well designed and the protocol is well-written. I have only one major concern that in the 2017 ACC annual meeting, the ABSORB-3 showed more MACE in the BVS arm at 2-year follow-up that warrants the safety in use of bioabsorbable stent for CAD patients, particularly those with small vessel diameter <2.5mm. Since those with small vessel diameter will not be excluded in the inclusion, whether 3-month DAPT treatment may be at risk for late TLF in these patients is unknown. In my opinion, the authors may need to provide more evidence to show the safety of 3-month DAPT using Orsiro and CX-ISAR in those with small vessel intervention.
---

REVIEWER	Chao-Lun Lai Department of Internal Medicine National Taiwan University Hospital Hsin-Chu Branch Taiwan
REVIEW RETURNED	27-Apr-2017

GENERAL COMMENTS	This study protocol describes a randomized controlled trial aiming at comparisons between two new generation drug-eluting stents and different durations of dual antiplatelet therapy. 1. Page 8, line 30. ".....whether short duration of DAPT is non-inferior to conventional 1 year duration". I suggest to add a statement "in patients receiving ultrathin newer generation drug-
--

	eluting stents” in the end of this sentence. 2. Page 9, line 52. Stenotic coronary lesion of > 50% diameter stenosis is defined as suitable for stent implantation in this trial. Why this protocol uses 50% stenosis as the criterion of significant stenosis rather than conventional 70%? Does this study use a central core angiographic laboratory to perform the quantitative coronary angiography? 3. Page 12, line 33. The trial applies permute-block randomization with a block size of 8. Does the trial apply stratified randomization using center (or hospital) as a stratum? 4. Page 14, how to deal with patients who are assigned to the 3-month DAPT arm but actually use DAPT for longer than 3 months? 5. Page 15, line 38. “Using the proportional-hazards model, clinical outcomes will be compared between the stent types and DAPT strategies, possibly after controlling for relevant covariates.” It is not necessary for a RCT to control for relevant covariates if randomization has been applied and well conducted. 6. Page 22, this study is also supported by Biotronik Korea and B. Braun Korea. Why this is not construed to be competing interests among the authors?
--	--

VERSION 1 – AUTHOR RESPONSE

Reviewer: 1

Reviewer Name: Gen-Min Lin

Institution and Country: Hualien-Armed Forces General Hospital, Taiwan

Reviewer’s comment:

Dr. Kim and colleagues will design a RCT, 2x2 factorial, non-inferior trial to compare TLF and DAPT between Orsiro and CX-ISAR.

In general, the study is well designed and the protocol is well-written.

I have only one major concern that in the 2017 ACC annual meeting, the ABSORB-3 showed more MACE in the BVS arm at 2-year follow-up that warrants the safety in use of bioabsorbable stent for CAD patients, particularly those with small vessel diameter <2.5mm. Since those with small vessel diameter will not be excluded in the inclusion, whether 3-month DAPT treatment may be at risk for late TLF in these patients is unknown. In my opinion, the authors may need to provide more evidence to show the safety of 3-month DAPT using Orsiro and CX-ISAR in those with small vessel intervention.

Response to reviewer #1 (Dr. Gen-Min Lin, Hualien-Armed Forces General Hospital, Taiwan).

Thank you for your careful review. As you pointed out, patients with small-vessel intervention are at a higher risk of experiencing ischemic adverse events and are generally recommended for prolonged DAPT. In fact, the intervention for small-vessel diameter (<3mm) is an item of the DAPT score and is included in the model to predict the occurrence of the future ischemic adverse events.¹ Recent guidelines recommend that the duration of DAPT maintenance be determined based on the patient's lesional and procedural characteristics.² For this reason, it is not surprising that patients with small-vessel intervention may have a poor outcome if they maintain DAPT for a shorter period of 3- or 6-months rather than the conventional 1-year DAPT.

However, the clinical efficacy of 3-month DAPT in small-vessel intervention has been demonstrated in several previous studies. In the RESET study that validated the clinical usefulness and safety of the 3-month DAPT, subgroup analysis was performed by dividing the diameter of the intervening vessels by 3 mm.³ Interestingly, even if the diameter of the vessel was small, maintaining DAPT for only 3 months did not adversely affect the clinical outcome (p-for-interaction 0.105). And the OPTIMIZE study, which was another randomized clinical trial using the 3-month DAPT regimen, showed no significant interaction with vessel size and duration of the DAPT maintenance (p-for-interaction

0.43).⁴ Both studies used the Endeavor and Resolute stents (zotarolimus-eluting stents), which are no longer used in clinical practice. Both the Orsiro or Coroflex-ISAR stents to be used in this HOST-IDEA trial have ultrathin strut thickness and minimized or absent polymer burden. As the stent platform is improved, we do not expect to find a significant difference in clinical outcomes even when the 3-month DAPT regimen is implemented.

In fact, both the Bioscience trial, a previous study using the Orsiro stent, and the ISAR-TEST 5 study, a trial using the previous version of the Coroflex-ISAR stent, showed that about one-third to half of the patients enrolled had a vessel size less than 2.75 mm.^{5 6} In these two studies, the clinical outcomes of the Orsiro and the prototype of the Coroflex-ISAR stents were not different from those of conventional everolimus- or zotarolimus-eluting stents even in small-vessel diameter. However, since these studies have applied the DAPT maintenance period of at least 6 months, there is no data to date on the 3-month DAPT maintenance. Based on these considerations, we also planned this study to prove that the combination of the 3-month DAPT regimen and the Orsiro or the Coroflex-ISAR stents might be clinically useful and safe for small-vessel intervention.

In order to reflect the comments of the reviewer and supplement the shortcomings, the following sentences were added to the last paragraph of the discussion:

In addition, this study may also provide meaningful data on the clinical usefulness of the 3-month DAPT regimen for small-vessel intervention. In fact, the intervention for small-vessel diameter (<3mm) is an item of the DAPT score and is included in the model to predict the occurrence of the future ischemic adverse events. In this regard, the recent guideline has advised that long-term DAPT maintenance should be considered for small-vessel intervention. There is, of course, data that contradict this recommendation. Some previous studies such as the RESET and OPTIMIZE trials have shown that the 3-month DAPT regimen is clinically useful and safe even in small-vessel intervention. However, these studies used the Endeavor and Resolute zotarolimus-eluting stents, which are no longer used in the current clinical practice. To date, detailed data on the effect of the combination of the 3-month DAPT regimen and the new stent platforms on small-vessel intervention are very scarce. In this context, even if the 3-month DAPT regimen were used for the Orsiro or the CX-ISAR stents, we expect that excellent clinical outcome can also be achieved in small-vessel intervention.

References

1. Yeh RW, Secemsky EA, Kereiakes DJ, et al. Development and Validation of a Prediction Rule for Benefit and Harm of Dual Antiplatelet Therapy Beyond 1 Year After Percutaneous Coronary Intervention. *JAMA* 2016;315(16):1735-49.
2. Bittl JA, Baber U, Bradley SM, et al. Duration of Dual Antiplatelet Therapy: A Systematic Review for the 2016 ACC/AHA Guideline Focused Update on Duration of Dual Antiplatelet Therapy in Patients With Coronary Artery Disease: A Report of the American College of Cardiology/American Heart Association Task Force on Clinical Practice Guidelines. *Journal of the American College of Cardiology* 2016;68(10):1116-39.
3. Kim BK, Hong MK, Shin DH, et al. A new strategy for discontinuation of dual antiplatelet therapy: the RESET Trial (REal Safety and Efficacy of 3-month dual antiplatelet Therapy following Endeavor zotarolimus-eluting stent implantation). *Journal of the American College of Cardiology* 2012;60(15):1340-8.
4. Feres F, Costa RA, Abizaid A, et al. Three vs twelve months of dual antiplatelet therapy after zotarolimus-eluting stents: the OPTIMIZE randomized trial. *JAMA* 2013;310(23):2510-22.
5. Pilgrim T, Heg D, Roffi M, et al. Ultrathin strut biodegradable polymer sirolimus-eluting stent versus durable polymer everolimus-eluting stent for percutaneous coronary revascularisation (BIOSCIENCE): a randomised, single-blind, non-inferiority trial. *Lancet* 2014;384(9960):2111-22.
6. Massberg S, Byrne RA, Kastrati A, et al. Polymer-free sirolimus- and probucol-eluting versus new generation zotarolimus-eluting stents in coronary artery disease: the Intracoronary Stenting and Angiographic Results: Test Efficacy of Sirolimus- and Probucol-Eluting versus Zotarolimus-eluting Stents (ISAR-TEST 5) trial. *Circulation* 2011;124(5):624-32.

Reviewer: 2

Reviewer Name: Chao-Lun Lai

Institution and Country: Department of Internal Medicine, National Taiwan University Hospital Hsin-Chu Branch, Taiwan

Reviewer's comments:

This study protocol describes a randomized controlled trial aiming at comparisons between two new generation drug-eluting stents and different durations of dual antiplatelet therapy.

1. Page 8, line 30. ".....whether short duration of DAPT is non-inferior to conventional 1 year duration". I suggest to add a statement "in patients receiving ultrathin newer generation drug-eluting stents" in the end of this sentence.

Response to reviewer #2, comment #1 (Dr. Chao-Lun Lai, Department of Internal Medicine, National Taiwan University Hospital Hsin-Chu Branch, Taiwan)

Thank you for your comment. This is a good suggestion. As you suggested, we added the statement at the end of the sentence.

2. Page 9, line 52. Stenotic coronary lesion of > 50% diameter stenosis is defined as suitable for stent implantation in this trial. Why this protocol uses 50% stenosis as the criterion of significant stenosis rather than conventional 70%? Does this study use a central core angiographic laboratory to perform the quantitative coronary angiography?

Response to comment #2:

This is a sharp point. This protocol reflects the real-world practice. The necessity for the intervention is decided at the operator's discretion (Therefore, we do not run a central core angiographic laboratory). Within our knowledge, the intermediate lesion is defined as a lesion with diameter stenosis of 50-70%. For example, FFR is indicated for those lesions with 50-70% stenosis to determine the necessity for the intervention. In other words, a stenotic lesion of > 50% diameter stenosis can be a candidate for stent implantation in real-world. Based on this situation, we used 50% stenosis as the criterion of significant stenosis.

3. Page 12, line 33. The trial applies permute-block randomization with a block size of 8. Does the trial apply stratified randomization using center (or hospital) as a stratum?

Response to comment #3:

Thank you for your comment. Our study randomly assigns patients using 'block randomization' with a block size of 8 patients. As mentioned in the methodology, this block size is designed exclusively for the purpose of determining the duration of the DAPT maintenance and allocating the stents for PCI (2x2 randomization study). Therefore, we did not use stratified assignment for the individual center. We have deliberately set a small block size to ensure more reliable random assignment. To make this more clear to readers, we added the following sentence to the manuscript:

"To ensure the randomization more secure, we have set a small block size for this study, and have not stratified by each participating center."

4. Page 14, how to deal with patients who are assigned to the 3-month DAPT arm but actually use DAPT for longer than 3 months?

Response to comment #4:

Because of the ischemic or bleeding events, even if the DAPT maintenance regimen is randomly assigned in advance at the registration stage, it is inevitable that some patients may not be able to complete the assigned maintenance period. This violation or drop-out may be caused by the patients' own decision or by choice of the medical staff. Therefore, as already mentioned in the method part (statistical analysis) of the manuscript, we will analyze the whole registered patient with both the

intention-to-treat analysis and the per-protocol analysis. Patients taking DAPT longer than three months will be excluded from the per-protocol analysis. The protocol violation is not expected to be that many, and the study was designed to enroll a sufficient number of patients in consideration of drop-out or protocol violation rates.

5. Page 15, line 38. "Using the proportional-hazards model, clinical outcomes will be compared between the stent types and DAPT strategies, possibly after controlling for relevant covariates." It is not necessary for a RCT to control for relevant covariates if randomization has been applied and well conducted.

Response to comment #5:

Because more than 2000 patients will be enrolled in this study, the clinical characteristics of each randomly assigned group are expected to be well balanced. However, as you pointed out in comment #4, protocol violation can occur, and the rates may differ slightly from group to group. Therefore, it is not possible to exclude that the baseline clinical characteristics of each patient group may be somewhat different. In that case, we will analyze the difference in clinical outcome according to the stent type and DAPT maintenance duration by controlling the influence of some unbalanced variables.

6. Page 22, this study is also supported by Biotronik Korea and B. Braun Korea. Why this is not construed to be competing interests among the authors?

Response to comment #6:

This study is an investigator-initiated clinical trial. At the time of the funding contract, the stent companies signed a contract stating that they would not interfere or control the investigators when conducting the study. In addition, the results of this research will be summarized and published independently so as not to advocate the economic benefits of the two stent companies that provided the funding. And there is no economic incentive provided to researchers. There is no additional reward for individual centers according to the number of registered patients, nor is there a separate registration goal allocated for each institution participating in the study. For this reason, we have stated that all the investigators do not have competing interests with the two stent companies in the protocol paper. Nevertheless, if the reviewer or editor points out that competing interest may arise because of financial support from the funding body, we will add relevant information to our manuscript.

We hope that our revised manuscript will be interesting and informative for your readers.

VERSION 2 – REVIEW

REVIEWER	Gen-Min Lin Hualien-Armed Forces General Hospital, Taiwan
REVIEW RETURNED	29-May-2017

GENERAL COMMENTS	The authors have made an adequate revision. I would like to accept the paper at the current stage.
---

REVIEWER	Chao-Lun Lai Department of Internal Medicine National Taiwan University Hospital Hsin-Chu Branch Hsin-Chu, Taiwan
REVIEW RETURNED	03-Jun-2017

GENERAL COMMENTS	I have no more comments.
--------------------------